# Linking mountaintop removal mining to water quality for imperiled species using satellite data

**Michael J. Evans**[1,2]*, **Kathryn Kay**[1º], **Chelsea Proctor**[1º], **Christian J. Thomas**[3º], **Jacob W. Malcom**[1,2º]

**1** Center for Conservation Innovation, Defenders of Wildlife, Washington, DC, United States of America,
**2** Environmental Science and Policy Dept., George Mason University, Fairfax, VA, United States of America,
**3** SkyTruth, Shepherdstown, WV, United States of America

º These authors contributed equally to this work.
* mevans@defenders.org

## Abstract

Environmental laws need sound data to protect species and ecosystems. In 1996, a proliferation of mountaintop removal coal mines in a region home to over 50 federally protected species was approved under the Endangered Species Act. Although this type of mining can degrade terrestrial and aquatic habitats, the available data and tools limited the ability to analyze spatially extensive, aggregate effects of such a program. We used two large, public datasets to quantify the relationship between mountaintop removal coal mining and water quality measures important to the survival of imperiled species at a landscape scale across Kentucky, Tennessee, Virginia, and West Virginia. We combined an annual map of the extent of surface mines in this region from 1985 to 2015 generated from Landsat satellite imagery with public water quality data collected over the same time period from 4,260 monitoring stations within the same area. The water quality data show that chronic and acute thresholds for levels of aluminum, arsenic, cadmium, conductivity, copper, lead, manganese, mercury, pH, selenium, and zinc safe for aquatic life were exceeded thousands of times between 1985 and 2015 in streams that are important to the survival and recovery of species on the Endangered Species List. Linear mixed models showed that levels of manganese, sulfate, sulfur, total dissolved solids, total suspended solids, and zinc increased by 6.73E+01 to 6.87E+05 µg/L and conductivity by 3.30E+06 µS /cm for one percent increase in the mined proportion of the area draining into a monitoring station. The proportion of a drainage area that was mined also increased the likelihood that chronic thresholds for copper, lead, and zinc required to sustain aquatic life were exceeded. Finally, the proportion of a watershed that was mined was positively related to the likelihood that a waterway would be designated as impaired under the Clean Water Act. Together these results demonstrate that the extent of mountaintop removal mining, which can be derived from public satellite data, is predictive of water quality measures important to imperiled species—effects that must be considered under environmental law. These findings and the public data used in our analyses are pertinent to ongoing re-evaluations of the effects of current mine permitting regulations to the recovery and survival of federally protected species.

**Data Availability Statement:** The data underlying the results presented in the study are available through an Open Science Framework repository: https://doi.org/10.17605/OSF.IO/A2Z34.

**Funding:** The authors received no specific funding for this work.

**Competing interests:** The authors have declared that no competing interests exist.

## Introduction

Natural ecosystems continue to degrade despite substantial global conservation efforts, in large part due to synergistic and large-scale impacts of human activities [1, 2]. For environmental laws to effectively protect natural ecosystems, a complete picture of the direct, indirect, and cumulative effects of potentially damaging, regulated actions needs to be analyzed and accounted for during the permitting process. While many direct, acute impacts of environmentally regulated actions are easy to measure, the myriad indirect and aggregate effects of human activities may require additional data or new techniques to uncover. For example, surface coal mining is regulated in many countries and directly impacts forest ecosystems through land clearing [3], which is easy to measure. However, surface mining also degrades surrounding forest by increasing fragmentation contributing to edge effects [4] and facilitating increased levels of non-native vegetation by compacting soil and altering its chemistry [5, 6]. Without consideration of these indirect, aggregate effects, environmental laws and policies will fail in their mission to protect the natural ecosystems upon which biodiversity and human life depend [7]. This degradation may be further underestimated if the disturbance of individual mines is evaluated in isolation without considering the accumulative effects of present, or future mines. Thus, data and approaches that can accommodate potentially complicated dynamics at large scales are needed for decision makers to come to better choices and meet their legal requirements [8, 9].

Mountaintop removal mining with valley fill (MTMVF) can have particularly extensive negative effects on environmental quality at large, landscape scales. MTMVF is a coal extraction practice in which coal seams in rugged terrain are accessed by first clearing overlying forest and then using explosives to remove overlying soil and bedrock. The leftover rock is then deposited into headwater stream valleys [10, 11], affecting aquatic and terrestrial ecosystems well beyond the immediate footprint of a mine. The conversion from forested land to bare earth decreases water quality, and streams linked to mine sites have higher rates of erosion and nutrient pollution and decreased habitat quality for aquatic species [5, 6, 11]. The release of alkaline mine drainage from weathering of rock and site waste at mine sites elevates conductivity and concentrations of metallic ions that negatively impact aquatic biota [12], leading to decreases in aquatic biodiversity [10, 13]. Additionally, pollutants can be transferred across food webs to terrestrial ecosystems [14].

The potential for MTMVF to negatively impact biodiversity through these interrelated direct and indirect effects is exacerbated by the massive spatial scale and extent of the practice. In the United States, MTMVF has been particularly prominent across Central Appalachia–a region encompassing the Appalachian Mountains in Kentucky, Tennessee, Virginia, and West Virginia—such that MTMVF is the primary cause of land cover change in the region [15]. Over 5,900 km$^2$ of Central Appalachian forest has been cleared for mining or mining related activities [16]. This level of landscape modification could pose a substantial threat to biodiversity anywhere, and in Central Appalachia the impacts of MTMVF are of particular concern because the region is a hot spot for endemic imperiled species [5]. Over 50 species on the endangered species list are found in the region. Many of these species are aquatic (e.g., darters, salamanders, and crayfish), making them particularly vulnerable to the negative water quality impacts of MTMVF.

In the United States, two critical environmental laws protecting aquatic habitats and species are the Clean Water Act (CWA) and the Endangered Species Act (ESA). Enacted in 1972, the CWA requires states to develop water quality standards for waterways to meet different usage criteria, including suitability for aquatic life, and restricts the discharge of pollutants into waterways not meeting these criteria [17]. The ESA focuses on the status of species and

requires federal agencies to ensure that their actions do not jeopardize the existence of species listed on the endangered species list (hereafter 'listed species') and do not adversely modify their designated critical habitat [18]. In terrestrial and freshwater contexts, federal agencies must consult with the US Fish and Wildlife Service (hereafter "the Service") on the effects of proposed projects and regulations on listed species [19]. In Oct. 2020, the Service completed a re-initiated consultation on the Surface Mining Control and Reclamation Act (SMCRA) [20], which defines regulations and procedures by which state regulatory authorities can issue mining permits. As with the original consultation on SMCRA in 1996, the Service found these procedures sufficient to prevent jeopardization of any listed species potentially affected by surface mining [20, 21] These conclusions were based on the requirement of permit seekers to analyze and present plans to minimize adverse effects to listed species. While spatially extensive analyses were considered impractical in 1996, new data and technology now make landscape-scale assessments possible.

In this paper, we use large public datasets and remote sensing analyses to quantify the potential indirect and aggregate impacts of MTMVF on imperiled species in Central Appalachia. The purpose of this study was to assess the impacts of MTMVF on water quality through the lens of protections to federally protected species required under the ESA. We evaluate whether statistical relationships exist between mountaintop removal mining activities as determined from satellite imagery and downstream water quality as measured at thousands of monitoring stations across the region. If downstream water quality can be predicted from remotely sensed, landscape-scale data of mining impacts, then improved estimates of baseline conditions and predictions of future conditions for imperiled species can be made. In turn, those results would be used to ensure regulatory decisions, such as those made under the ESA, are meeting the purposes of the law. To that end, we evaluated the relationships between mined area and:

1. observed values of water quality measures relevant to aquatic life;

2. the frequency that water quality thresholds for aquatic life were exceeded; and

3. the frequency of waterway impairment under the Clean Water Act.

We demonstrate that such relationships often exist and link them to the specific laws and policies that provide requirements and mechanisms to ameliorate them. We focus on variables that are known or believed to be important for aquatic species persistence to provide insights that can be used to directly improve the conservation prospects for federally protected species.

## Methods

### Mining data

We obtained spatial data delineating the footprints of all large surface mines across Central Appalachia in each year from 1985 to 2015. These data were generated using Landsat satellite imagery in a previous analysis measuring trends in the extent of mining activities over time [16]. In order to avoid commission errors in mine identification affecting subsequent analyses, we cross-referenced these footprints with each year of available data from the National Land Cover Dataset [22] and eliminated all footprints that overlapped with areas flagged as agriculture or development. We used these mine footprints to define our study area as a contiguous selection of all the US counties containing these mines (Fig 1).

### Water quality data

We obtained measurements of water quality from the national water quality data portal [23] using the *dataRetrieval* package [24] for R [25]. The national water quality data portal

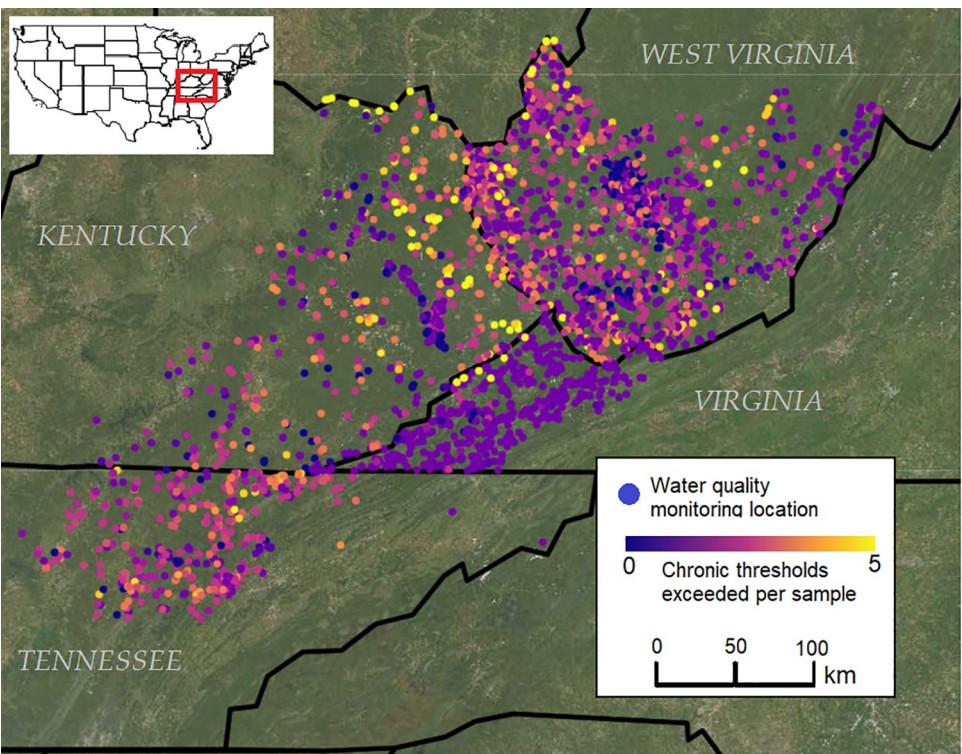

**Fig 1. Chronic exposure thresholds for aquatic life were exceeded thousands of times in Central Appalachian waterways between 1985 and 2015.** Map shows the locations of water quality monitoring stations within the study area encompassing parts of Kentucky, Tennessee, Virginia, and West Virginia shown on the inset map. Colors indicate the mean number of different water quality measures for which chronic exposure thresholds were exceeded each time a sample was taken at a given location. The basemap contains USGS/NASA Landsat data from 2020, accessed through Google Earth Engine.

aggregates data from monitoring stations nationally, primarily from the U.S. Geological Survey and the Environmental Protection Agency, but also the National Park Service and other state agencies. We selected a set of water quality measures related to mining activity that can also affect the health of aquatic species (Table 1). Additionally, we collected flow rate and temperature data, although these measures were not ultimately used. Water quality data from 1985 through 2015 were collected—corresponding to the same period over which mining footprint data was available—from all monitoring stations within the counties comprising our study area. Water quality data were provided in different units, and we standardized all measures of concentration to ug/L, conductivity to μS /cm, and turbidity to NTU. To account for potentially mis-recorded data, we flagged as outliers any observations above the 99.9th percentile for a given measure. This threshold was determined empirically by plotting the number of observations falling outside successively larger quantiles and selecting the quantile at which observations appeared to plateau.

To identify acute and chronic toxicity thresholds for different water quality measures, we used the state water quality standards for aquatic life administered by Virginia under the Clean Water Act [1]. These standards are approved by the Environmental Protection Agency and are used to determine waterway impairment requiring mitigation. The water quality standards thus represent an agreed upon set of thresholds necessary to maintain suitability of waterways for aquatic species. The Virginia standards were identical to those from Kentucky, Tennessee, and West Virginia. For each measure we flagged any observation where recorded levels

**Table 1. Chronic and acute toxicity thresholds were exceeded many times in waterways important to listed aquatic species.**

| Measure | Chronic | Acute | No. sites (%) |
|---|---:|---:|---:|
| Aluminum | 3,853 | 871 | 973 (51.6) |
| Arsenic | 39 | 39 | 831 (44.0) |
| Cadmium | 659 | 629 | 853 (45.2) |
| Calcium[a] | - | - | - |
| Conductivity | 2,246 | 2,244 | 226 (12.0) |
| Copper | 1,052 | 750 | 927 (49.1) |
| Iron[a] | - | - | - |
| Lead | 795 | 271 | 878 (46.5) |
| Manganese | 3,651 | 2,276 | 1,073 (56.9) |
| Mercury | 72 | 62 | 722 (38.3) |
| pH | 15,419 | 15,419 | 1,187 (62.9) |
| Selenium | 113 | 79 | 793 (42.0) |
| Sulfate[a] | - | - | - |
| Turbidity[a] | - | - | - |
| Total dissolved solids[a] | - | - | - |
| Total suspended solids[a] | - | - | - |
| Zinc | 451 | 451 | 997 (52.8) |

[a]No chronic or acute thresholds provided for measure.

Table shows the number of times that any recorded value (e.g. 'Dissolved', 'Total', etc.) of each water quality measure exceeded standard thresholds for aquatic life, and the number of different monitoring stations at which these events occurred. These data only consider measures taken from 1,887 monitoring stations whose drainage basin contained a stream that was designated as important to species survival and recovery.

exceeded either the acute or chronic exposure thresholds. Finally, we also obtained the locations of all waterways declared as 'impaired' under section 303(d) of the Clean Water Act occurring within our study area from the EPA's Environmental Dataset Gateway [26]. Impaired waterway designations were made on biannual cycles beginning in 1991.

## Imperiled species data

We compiled a list of ESA-listed aquatic species whose range overlapped the study area (S1 Table) by using the ECOS Data Explorer (https://ecos.fws.gov/ecp/report/adhocDocumentation?catalogId=species&reportId=species). Potentially important water quality measures were determined by examining federal documents pertaining to these species including listing decisions, recovery plans, and five-year reviews (available at https://ecos.fws.gov/ecp). We generated a list of streams that were important to the survival and recovery of listed aquatic species as those streams identified in these same documents as containing extant populations or as being necessary for recovery. For species with designated critical habitat, we included these waterways as well. We refer to this combined list as streams important to imperiled species.

## Spatial analyses

To assess the relationship between mining and water quality measures recorded at monitoring stations, we associated mines and monitoring locations based on hydrography. We used the watershed and flow modeling tools from the *pyshed* package [27] for Python to delineate the geographic areas that drained into each monitoring station. These analyses required a model

of surface elevation, and we used a 30m digital elevation model provided by NASA [28] clipped to our study area. Once drainage basins were delineated for each monitoring station, we calculated the proportion of each basin covered by surface mines in each year from 1985 to 2015. We also created a many-to-one spatial join identifying which mines fell within the drainage basin of each monitoring station.

We used the annual Cropland Data Layers [29] and NLCD data to estimate the proportion of drainage basins that were covered by agriculture or impervious surface in each year for which mining footprint data was available (1985–2015). Cropland data were available annually beginning in 2009, and NLCD impervious surface data were available from 1992, 2001, 2004, 2006, 2008, 2011 and 2013. We interpolated between and extrapolated beyond these observed data points to estimate agricultural and impervious surface data for missing years between 1985 and 2015.

In analyses pertaining to 303(d) impaired waterways we consider mined area within watersheds containing the waterway. We used U.S. Geological Survey HUC12 hydrologic units [30] to represent watersheds. We then repeated the above interpolation and extrapolation procedure to obtain the area mined and covered by agriculture or impervious surface within a contiguous selection of watersheds overlapping the mine footprint data set.

Unless otherwise indicated, all spatial analyses were performed using the Google Earth Engine python API [31].

## Statistical analyses

We tallied the frequency with which water quality standards were exceeded in streams important to imperiled species by spatially joining the locations of monitoring stations to linear stream features with attributes indicating whether the stream was important to imperiled species.

In all analyses estimating the relationship between mined area and water quality measures, we attempted to account for attenuation in pollutant concentrations with increasing distance between monitoring stations and mines. We adjusted the area of each mine footprint within a given drainage basin in proportion to the square root of the distance from the mine to the corresponding monitoring station. We refer to this measure as adjusted mined area.

Our first objective was to quantify the relationship between values of each water quality measure and the proportion of drainage basins that were mined each year, while controlling for agriculture and impervious surface in the drainage basin. To do so, we created linear mixed effects models with normal error distributions and random intercepts per year nested within monitoring sites. These models were used to estimate the increase or decrease in mean water quality measures as a function of adjusted mined area, percent agriculture, and percent impervious surface within drainage basins.

Our second objective was to determine whether the proportion of a drainage basin that was mined each year affected the probability that pollutant levels would exceed thresholds deemed safe for aquatic life. We specified a generalized linear mixed model using a binomial error distribution and logit link, with random intercepts per year nested within monitoring sites. These models were used to predict the probability that an observed value for a given measure would exceed chronic thresholds as a function of adjusted mined area, percent agriculture, and percent impervious surface within drainage basins.

Our last objective was to estimate the relationship between the proportion of a watershed that was mined and the probability that a waterway therein would be designated as impaired based on the standards for aquatic life under the Clean Water Act. We specified a generalized linear mixed model with a logit link and binomial error distribution with random intercepts

per year nested within monitoring stations. These models were used to predict the probability that a watershed would contain an impaired waterway at the end of a given biannual evaluation cycle as a function of adjusted mined area, percent agriculture, and percent impervious. Because impaired waterways are only tallied during biannual cycles, we used the maximum percent agriculture and impervious surface within two-year cycles as predictor variables. To account for lagged and accumulative effects, we used the cumulative sum of percent mined area within drainage basins over time.

In all regression analyses we included only sites with at least 10 observations. We estimated model parameters in a Bayesian framework using the *rstanarm* [32] package for R. For each model we generated four MCMC chains and tested for convergence using the Rhat statistic. A significant relationship was determined between mined area and response variables if the 95% credible interval around the relevant parameter estimate did not overlap zero. We tested for collinearity among the predictor variables used in linear models (percent mined area, percent agriculture, percent impervious surface) using pairwise correlation coefficients. Code used in all analyses is available through an Open Science Framework repository [33].

## Results

We obtained water quality data from 4,260 different water quality monitoring sites across our study area. Distances between stations and mines were exponentially distributed, ranging from < 1 to 343 km (x = 77.7 km, $\sigma^2$ = 4,071 km). The number of observations (i.e., occasions on which water quality data was recorded) at each site ranged from 1 to 275. There were 569 sites with at least 10 observations and drainage basin areas greater than 400 km$^2$ that were included in modeling analyses. None of the predictor variables exhibited evidence of collinearity (-0.09 < $R^2$ < 0.07).

Linear mixed models indicated significant positive relationships between the proportion of a drainage basin that was mined and measured levels of conductivity, manganese, sulfate, sulfur, total dissolved solids, total suspended solids, and zinc (Fig 2). No measures were significantly negatively associated with adjusted mined area (Table 2).

We also found significant positive relationships between the cumulative proportion of a watershed that was mined over time and the log odds that a stream in that watershed would be designated as impaired for aquatic life under the Clean Water Act (Table 3).

Measures of conductivity all exceeded chronic exposure thresholds, and we were unable to model the probability of exceeding thresholds as a function of land cover predictors. Models indicated that the probabilities that chronic exposure thresholds for copper, lead, and manganese were exceeded were all positively related to the area mined within drainage basins (Fig 3). The probabilities for no water quality measures exhibited a significant negative relationship.

As of 2018, 55 ESA-listed aquatic species potentially occurred within the counties comprising our study area. These included 39 mollusk, 12 fish, 3 crustacean, and 1 snail species. Of these listed species, 16 had designated critical habitat (S1 Table). Additionally, for 50 of these species we were able to identify specific streams that were important to the species survival and recovery in either critical habitat designations, listing decisions, five-year reviews, or recovery plans. Of the 4,260 monitoring stations, 2,881 of these drained areas containing important streams. Chronic and acute toxicity thresholds for aquatic life were exceeded thousands of times at these monitoring stations (Table 1). The most frequently exceeded threshold was that for pH, followed by manganese, aluminum, and conductivity (Table 1). 54 streams designated as critical habitat were sampled directly by 209 monitoring stations. Water quality thresholds were exceeded at least once at each of those monitoring stations a total of 5,592 times with a maximum of 272 at a single station.

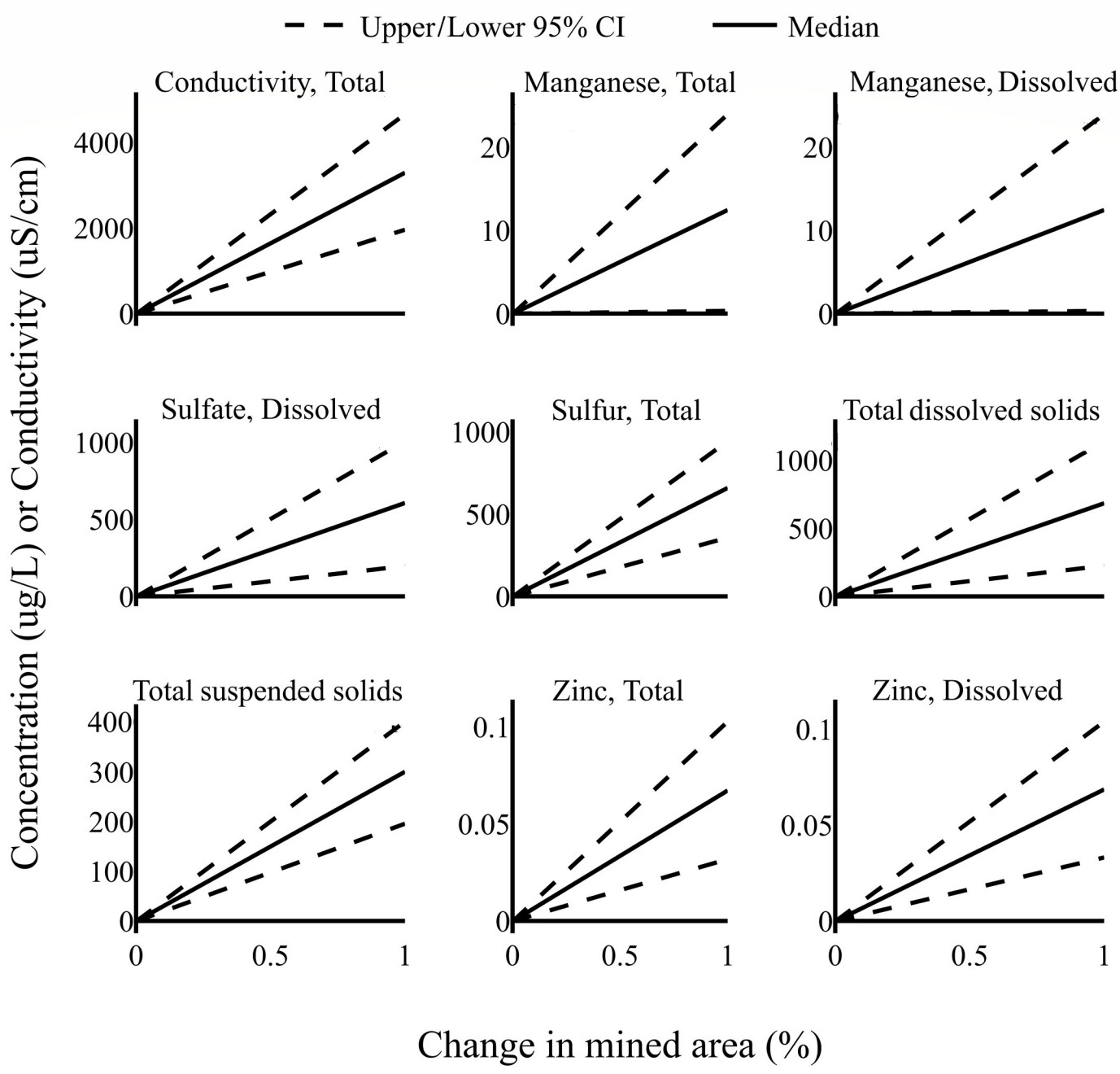

**Fig 2. Increases in the proportion of drainage basins that were mined lead to increases in multiple measures of water quality that are detrimental to aquatic species.** Graphs show the change in water quality measures per change in mined area as estimated by linear mixed models. Dashed lines encompass a 95% credible interval around estimated relationships.

## Discussion

The scale of the human enterprise is large and growing [7]. To minimize the environmental impacts of human activities, analytical approaches that can address the full scale and complexity of their effects are required. Many laws and regulations are theoretically capable of handling such large-scale effects, but too often the data and science needed to fully understand the

**Table 2. Significant positive relationships were estimated between mined area and nine measures of water quality.**

| Toxin | Measure | Median | 2.5% | 97.5% | Rhat |
|---|---|---|---|---|---|
| Aluminum (μg/L) | Total | 3.57E+03 | -1.01E+03 | 8.17E+03 | 1.01 |
| Aluminum (μg/L) | Dissolved | 3.74E+03 | -9.82E+02 | 8.52E+03 | 1.01 |
| Arsenic (μg/L) | Total | -5.03E+01 | -1.38E+02 | 3.52E+01 | 1.00 |
| Arsenic (μg/L) | Dissolved | 3.39E-01 | -5.26E+00 | 5.70E+00 | 1.00 |
| Cadmium (μg/L) | Total | -8.21E+02 | -2.08E+03 | 4.52E+02 | 1.01 |
| Cadmium (μg/L) | Dissolved | -6.92E+00 | -4.39E+01 | 3.11E+01 | 1.00 |
| Calcium (μg/L) | Total | 6.34E+04 | -4.87E+02 | 1.25E+05 | 1.01 |
| Calcium (μg/L) | Dissolved | 8.04E+04 | -3.35E+04 | 1.92E+05 | 1.00 |
| **Conductivity (μS/L)** | **Total** | **3.30E+06** | **1.96E+06** | **4.67E+06** | **1.00** |
| Copper (μg/L) | Total | -1.20E+02 | -4.52E+02 | 2.03E+02 | 1.00 |
| Copper (μg/L) | Dissolved | -1.26E+02 | -4.67E+02 | 2.04E+02 | 1.00 |
| Iron (μg/L) | Total | -2.04E+03 | -2.23E+04 | 1.87E+04 | 1.00 |
| Iron (μg/L) | Dissolved | -1.44E+04 | -3.48E+04 | 5.94E+03 | 1.00 |
| Lead (μg/L) | Total | -5.12E+01 | -2.32E+02 | 1.28E+02 | 1.00 |
| Lead (μg/L) | Dissolved | -5.32E+01 | -2.27E+02 | 1.16E+02 | 1.00 |
| **Manganese (μg/L)** | **Total** | **1.24E+04** | **3.25E+02** | **2.40E+04** | **1.01** |
| **Manganese (μg/L)** | **Dissolved** | **1.24E+04** | **3.25E+02** | **2.40E+04** | **1.01** |
| Mercury (μg/L) | Total | -5.52E+00 | -3.42E+01 | 2.33E+01 | 1.00 |
| Mercury (μg/L) | Dissolved | -5.77E-01 | -2.82E+00 | 1.76E+00 | 1.01 |
| pH | Dissolved | 2.68E-01 | -3.37E-01 | 9.02E-01 | 1.00 |
| Selenium (μg/L) | Total | -1.78E+02 | -4.01E+02 | 4.65E+01 | 1.00 |
| Selenium (μg/L) | Dissolved | -1.76E+02 | -4.10E+02 | 5.36E+01 | 1.00 |
| **Sulfate (μg/L)** | **Dissolved** | **6.08E+05** | **1.98E+05** | **1.01E+06** | **1.00** |
| **Sulfur (μg/L)** | **Total** | **6.58E+05** | **3.58E+05** | **9.45E+05** | **1.00** |
| **Total dissolved solids (μg/L)** | **Total** | **6.87E+05** | **2.25E+05** | **1.14E+06** | **1.00** |
| Total dissolved solids (μg/L) | Dissolved | 7.74E+05 | -3.56E+04 | 1.55E+06 | 1.00 |
| **Total suspended solids (μg/L)** | **Total** | **3.01E+05** | **1.96E+05** | **4.02E+05** | **1.00** |
| Turbidity (NTU) | Total | -7.06E+02 | -2.79E+03 | 1.33E+03 | 1.00 |
| **Zinc (μg/L)** | **Total** | **6.73E+01** | **3.19E+01** | **1.03E+02** | **1.01** |
| **Zinc (μg/L)** | **Dissolved** | **6.85E+01** | **3.32E+01** | **1.04E+02** | **1.01** |

Table shows the regression coefficients quantifying the relationship between adjusted mined area on measures, as estimated by linear mixed models. The 50th, 2.5th, and 97.5th percentile of the posterior distribution, as well as a measure of MCMC chain convergence (Rhat) are included. Bold text indicates estimates with 95% credible intervals that did not overlap zero.

effects of our actions are limited [8]. Here we combine two classes of large, public datasets to demonstrate that mountaintop mining with valley fill (MTMVF) in Central Appalachia is associated with degraded water quality at landscape scales in ways that affect the survival and recovery of federally protected species. We found consistent evidence linking changes in mined area with increases in concentrations of toxics, conductivity, and dissolved and suspended solids. Far from being innocuous side effects, the measures with the strongest relationships to mining were among those that directly affect the survival of aquatic species [12]. These findings demand regulatory action under federal environmental laws including the Endangered Species Act and Clean Water Act, because these activities are concentrated in an area with high numbers of imperiled species.

**Table 3. Significant positive relationships were estimated between mined area in a watershed and the probability that streams therein were designated as impaired.**

| Max percentage | Median | 2.5% | 97.5% | Rhat |
|---|---|---|---|---|
| Mined | 23.04 | 20.34 | 25.88 | 1.00 |
| Impervious | 97,232.78 | 74,617.36 | 118,089.30 | 1.02 |
| Cultivated | -36,071.50 | -40,454.80 | -32,007.00 | 1.00 |

Table shows the 50th, 2.5th, and 97.5th percentiles of posterior distributions of regression coefficients quantifying the relationship between and probability of impairment, as estimated by logistic regression models with random effects. Measures of MCMC chain convergence (Rhat) are included.

The positive relationships between the extent of mined areas and degradation in water quality that we identified were not surprising. A large body of previous ecological and hydrologic research has shown that surface mining can negatively impact water quality and reduce the suitability of streams for aquatic species at local and regional scales [3, 10, 11, 13, 14]. Consistent with this research, we found substantial increases in stream conductivity, and the concentrations of manganese, sulfate, sulfur, zinc, and dissolved and suspended solids associated with increases in the proportion of upstream areas that were mined. Our results build on previous work by illustrating that MTMVF not only degrades water quality immediately proximate to mines, it does so at a landscape scale—establishing a direct relationship between the aggregate area mined on the landscape and degradation in water quality and habitat suitability for aquatic species. In the context of environmental regulations governing the permitting and operation of MTMVF mines, these direct and aggregate effects illustrate the importance of considering a spatially extensive set of conditions when evaluating the environmental baseline and potential impacts to protected ecosystems and species.

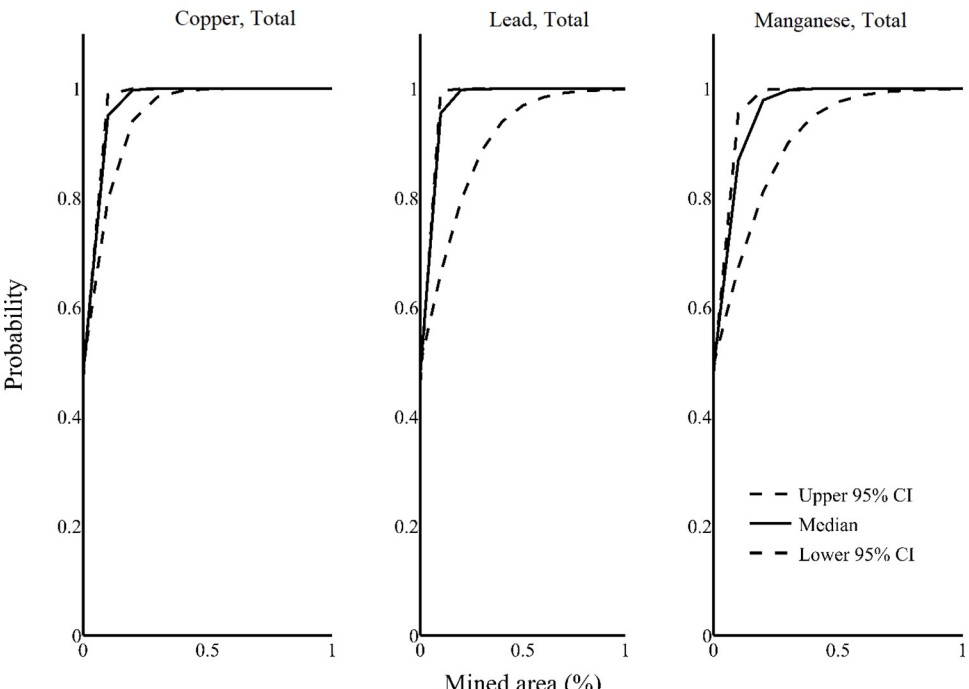

**Fig 3. Increases in the proportion of drainage basins that were mined increased the probability that chronic exposure toxicity thresholds would be exceeded for three water quality measures.** Graphs show the change in probability of exceedance per change in mined area as estimated by linear mixed models. Dashed lines encompass the 95% credible interval around estimated relationships.

Accounting for the aggregate effects of mine permitting at landscape scale is particularly important in the context of conserving species threatened with or on the brink of extinction, such as those on the endangered species list. We found that degradation in water quality progressed to potentially lethal levels for aquatic life—documenting thousands of instances in which chronic and acute water quality thresholds were exceeded between 1985 and 2015. Our analyses demonstrate a positive relationship between the probability that chronic thresholds were exceeded and the amount of upstream area that was mined. Measured levels of dissolved copper, lead, and manganese were more likely to exceed safe thresholds for aquatic life as the proportion of upstream area that was mined increased. Together, these dynamics likely contributed to the emergent outcome we observed that a waterway was more likely to fail to meet water quality standards and be declared impaired under the CWA as the proportion of its watershed that was mined increased.

Mining activities thus directly impacted listed species, based on the observation that water quality thresholds were exceeded thousands of times at monitoring stations on streams designated as critical habitat and stations draining streams identified by the US Fish and Wildlife Service as important to listed species survival and recovery. For instance, monitoring station 211WVOWR-KE-000-004.50 collected water from a section of the Elk River designated as critical habitat for the diamond darter (*Crystallaria cincotta*). Out of 48 occasions, chronic and acute thresholds for aluminum were exceeded 33 and 8 times, respectively; manganese 14 and 5 times; copper 2 and 2 times; and cadmium once. The repeated exceedance of acute toxicity thresholds for aquatic life within designated critical habitat could constitute an adverse modification of critical habitat–an outcome federal agencies must legally avoid under the ESA. The regular and repeated exceedance of water quality thresholds for aquatic life in both critical habitat, and waterways important to the survival and recovery of listed species would seem to conflict with the purpose of the ESA, protecting and recovering imperiled species.

Evidence that the extent of impacts of MTMVF can include entire watersheds should inform the implementation of federal environmental regulations. Regulations under SMCRA require mine operators to minimize the impact of their actions within 'adjacent areas,' which are defined as the areas within which imperiled species '. . .reasonably could be expected to be adversely impacted by proposed mining operations' [20]. That is, if the areas impacted by MTMVF extend into broader areas where ESA-listed species occur, those impacts must be considered in consultation [19]. Under SMCRA, mine operators must also minimize adverse impacts and enhance natural resources during mine reclamation. Reclamation activities are increasingly relevant for the protection and recovery of imperiled species as the use of coal for energy, and hence its production, has been declining in the United States and is expected to continue to decline [34]. Reclamation efforts have historically been unsuccessful re-establishing native Appalachian forests following MTMVF [15], as they often emphasize restoring vegetation to mined sites without prioritizing native species [35, 36]. Even efforts that successfully re-establish native communities often exclude rare species [37]. However, the Forestry Reclamation Approach provides management practices which allow operators to meet the reclamation requirements of SMCRA, while also restoring native forests [38]. Both the Forestry Reclamation Approach, and the data presented here constitute 'best available scientific and commercial information,' which must be considered under the ESA [18]. Whether their exclusion from the Service's past [20, 21] or future analyses of the effects of implementing SMCRA violates this requirement is an outstanding question beyond the scope of this paper, but may be a significant legal vulnerability.

Of course, water quality degradation and waterway impairment are not solely attributable to surface mining, and more than half of US waterways do not meet CWA water quality standards. In addition to point source pollution, many land use factors can contribute to the

degradation of aquatic conditions, including the prevalence of impervious surfaces and agriculture within watersheds [39, 40]. These pathways were reflected in our results, which show positive relationships between impervious surface and agricultural extent within watersheds and the likelihood that an encompassed waterway would be impaired. In recent years, however, energy has been the largest driver of land use change in the United States [41], and policies that minimize the negative environmental impacts of energy exploration, extraction, and production thus have the potential to make a significant improvement to the conservation prospects of biodiversity.

## Conclusions

Our results demonstrate that even after accounting for additional sources of water quality degradation, surface mining contributed to further increases in the concentration of toxics that can impair aquatic biota in Central Appalachia. These findings indicate that, in situ, the growth, continued operation, and legacy effects of MTMVF in the region likely directly limit the prospects for survival and recovery of over 50 federally protected species. Federal agencies will need to take at least two steps to meet their obligations under federal statutes given these results. First, existing and new critical habitat designations must be accurately accounted for by federal agencies as they implement programs such as SMCRA and carry out consultation on those programs under the ESA. Absent this, the ESA is at risk of being nothing more than a "paper tiger" in which protection only exists on paper rather than benefiting species [42]. Additionally, the requirements for adverse effect minimization and resource enhancement under SMCRA can use the same publicly available water quality and remote sensing data and analytical methods presented here to account for the effects of mine operation beyond the immediate footprint. While approved practices for assessing and permitting the operation of an individual mine may successfully mitigate impacts to immediately surrounding ecosystems, our results demonstrate that a landscape scale assessment is necessary to fully account for the impacts of surface mining on imperiled species.

## Supporting information

**S1 Table. 55 aquatic species that occur within Central Appalachia are listed on the Endangered Species List.** Table shows the common and scientific names of listed species, dates on which species were added to the Endangered Species List and their current status (Threatened or Endangered), and the date at which any critical habitat was designated. [a]Proposed critical habitat after completion of the study.
(DOCX)

## Acknowledgments

We thank Kat Diersen and Robert Dreher for review and revisions of the manuscript. Additionally, we thank the team at the Center for Conservation Innovation for providing continuous suggestions and input on the goals and methods of the analysis.

## Author Contributions

**Conceptualization:** Michael J. Evans, Jacob W. Malcom.

**Data curation:** Michael J. Evans, Kathryn Kay, Chelsea Proctor, Christian J. Thomas.

**Formal analysis:** Michael J. Evans, Chelsea Proctor.

**Investigation:** Kathryn Kay, Chelsea Proctor.

**Methodology:** Michael J. Evans.

**Project administration:** Michael J. Evans.

**Supervision:** Jacob W. Malcom.

**Visualization:** Michael J. Evans.

**Writing – original draft:** Michael J. Evans, Christian J. Thomas, Jacob W. Malcom.

**Writing – review & editing:** Michael J. Evans, Kathryn Kay, Chelsea Proctor, Christian J. Thomas, Jacob W. Malcom.

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
