## [Decision Letter · Decision Letter 0]

15 Jan 2021

PONE-D-20-28542

Mountaintop Removal Mining Threatens the Survival and Recovery of Imperiled Species

PLOS ONE

Dear Dr. Evans,

Thank you for submitting your manuscript to PLOS ONE. After careful consideration, we feel that it has merit but does not fully meet PLOS ONE’s publication criteria as it currently stands. Therefore, we invite you to submit a revised version of the manuscript that addresses the points raised during the review process.

REVIEWER I

Major comments

The introduction is too long and should be shortened and made more concise

There are also two major flaws with the manuscript

1). Analysis of biota in contaminated and uncontaminated areas is never explicitly compared, paired tests and analysis are needed to establish a causal impact. The manuscript repeatedly states that tolerances are being exceeded, but the impacts on biota are not clearly analysed. Much of the manuscript reads as if the initial aim was to look at mining and pollution, and that biodiversity was added later to increase readership and relevance

Many more explicit details on the impacts are needed.

Secondly, and following on this, better species specific analysis (i.e. by guild) are needed in relation to sensitivity, and contaminant specific analysis also needed. The main headline here is "mining contaminates rivers, this could be problematic" but explicit analysis on how species are impacted is needed. This will require extensive restructuring and further analysis to elaborate the findings of the paper.

REVIEWER II

The study provides important insights into how mining operations conducted upstream can have cascading effects downstream, degrading aquatic ecosystems and causing detrimental effects on biodiversity.

Following are some minor revisions suggested:

1. Introduction: In the first paragraph, the authors have reiterated about “detrimental effects of environmentally damaging activities”. I suggest the authors to provide a few examples of such damaging activities and their potential impacts to introduce the study in a more comprehensible manner.

2. Please rephrase the line – “Additionally, pollutants can be transferred through food webs to downstream terrestrial ecosystems (14)”. I think that the authors meant to convey that pollutants can be transferred downstream by water flow, across food webs, and into terrestrial ecosystems.

3. I suggest the authors to provide the year of enactment of the acts mentioned in the paper to provide non-native readers an idea about the timeline.

4. Were any correlation tests performed to rule out correlated variables from the GLMMs?

5. The authors should provide a table of the variables used in the study.

6. Figure 1: Please provide a map of the USA with the highlighted study area.

We look forward to receiving your revised manuscript.

Kind regards,

Shalini Dhyani, Ph.D

Academic Editor

PLOS ONE

Journal Requirements:

Thanks and Regards,

Shalini Dhyani

2. We noted in your submission details that a portion of your manuscript may have been presented or published elsewhere.

"Yes. The mine footprint data used to estimate was previously published in Pericak A et al. (2018) Mapping the yearly extent of surface coal mining in central appalachia using landsat and Google Earth Engine. PLoS One 13(7). These data are publicly available at: " ext-link-type="uri" xlink:type="simple">https://skytruth.org/mtr-data-files/"

Please clarify whether this publication was peer-reviewed and formally published. If this work was previously peer-reviewed and published, in the cover letter please provide the reason that this work does not constitute dual publication and should be included in the current manuscript.

3. We note that Figure 1 and Supplementary Figures F1, F2 in your submission contain map/satellite images which may be copyrighted. All PLOS content is published under the Creative Commons Attribution License (CC BY 4.0), which means that the manuscript, images, and Supporting Information files will be freely available online, and any third party is permitted to access, download, copy, distribute, and use these materials in any way, even commercially, with proper attribution. For these reasons, we cannot publish previously copyrighted maps or satellite images created using proprietary data, such as Google software (Google Maps, Street View, and Earth). For more information, see our copyright guidelines: http://journals.plos.org/plosone/s/licenses-and-copyright.

3.1.    You may seek permission from the original copyright holder of Figure 1 and Supplementary Figures F1, F2 to publish the content specifically under the CC BY 4.0 license. 

3.2.    If you are unable to obtain permission from the original copyright holder to publish these figures under the CC BY 4.0 license or if the copyright holder’s requirements are incompatible with the CC BY 4.0 license, please either i) remove the figure or ii) supply a replacement figure that complies with the CC BY 4.0 license. Please check copyright information on all replacement figures and update the figure caption with source information. If applicable, please specify in the figure caption text when a figure is similar but not identical to the original image and is therefore for illustrative purposes only.

Reviewers' comments:

Reviewer's Responses to Questions

**Comments to the Author**

1. Is the manuscript technically sound, and do the data support the conclusions?

Reviewer #1: Yes

Reviewer #2: Yes

2. Has the statistical analysis been performed appropriately and rigorously? 

Reviewer #1: Yes

Reviewer #2: Yes

3. Have the authors made all data underlying the findings in their manuscript fully available?

Reviewer #1: Yes

Reviewer #2: Yes

4. Is the manuscript presented in an intelligible fashion and written in standard English?

Reviewer #1: Yes

Reviewer #2: Yes

5. Review Comments to the Author

Reviewer #1: The study provides important insights into how mining operations conducted upstream can have cascading effects downstream, degrading aquatic ecosystems and causing detrimental effects on biodiversity. The authors can improve the manuscript by going through some minor changes suggested in this review.

Following are some minor revisions suggested:

1. Introduction: In the first paragraph, the authors have reiterated about “detrimental effects of environmentally damaging activities”. I suggest the authors to provide a few examples of such damaging activities and their potential impacts to introduce the study in a more comprehensible manner.

2. Please rephrase the line – “Additionally, pollutants can be transferred through food webs to downstream terrestrial ecosystems (14)”. I think that the authors meant to convey that pollutants can be transferred downstream by water flow, across food webs, and into terrestrial ecosystems.

3. I suggest the authors to provide the year of enactment of the acts mentioned in the paper to provide non-native readers an idea about the timeline.

4. Were any correlation tests performed to rule out correlated variables from the GLMMs?

5. The authors should provide a table of the variables used in the study.

6. Figure 1: Please provide a map of the USA with the highlighted study area.

Reviewer #2: The introduction is too long and should be shortened and made more concise

There are also two major flaws with the manuscript

1). Analysis of biota in contaminated and uncontaminated areas is never explicitly compared, paired tests and analysis are needed to establish a causal impact. The manuscript repeatedly states that tolerances are being exceeded, but the impacts on biota are not clearly analysed. Much of the manuscript reads as if the initial aim was to look at mining and pollution, and that biodiversity was added later to increase readership and relevance

Many more explicit details on the impacts are needed

Secondly, and following on this, better species specific analysis (i.e. by guild) are needed in relation to sensitivity, and contaminant specific analysis also needed. The main headline here is "mining contaminates rivers, this could be problematic" but explicit analysis on how species are impacted is needed. This will require extensive restructuring and further analysis to elaborate the findings of the paper

6. PLOS authors have the option to publish the peer review history of their article (what does this mean?). If published, this will include your full peer review and any attached files.

Reviewer #1: **Yes: **Syed Ainul Hussain, Ph.D., D.Sc.

Wildlife Institute of India, Dehradun, India

Reviewer #2: No

---

## [Author Response · Author response to Decision Letter 0]

19 Mar 2021

REVIEWER 1

Comment 1: Introduction: In the first paragraph, the authors have reiterated about “detrimental effects of environmentally damaging activities”. I suggest the authors to provide a few examples of such damaging activities and their potential impacts to introduce the study in a more comprehensible manner. 

Response: Thank you for the suggestion; we have incorporated examples of the ways in which surface mining degrades surrounding forested ecosystems, previously in the second Introduction paragraph, into this first paragraph to provide a comprehensible example of these dynamics. 

Comment 2: Please rephrase the line – “Additionally, pollutants can be transferred through food webs to downstream terrestrial ecosystems (14)”. I think that the authors meant to convey that pollutants can be transferred downstream by water flow, across food webs, and into terrestrial ecosystems.

Response: We have revised the sentence. It now reads "Additionally, pollutants can be transferred across food webs to terrestrial ecosystems."

Comment 3: I suggest the authors to provide the year of enactment of the acts mentioned in the paper to provide non-native readers an idea about the timeline. 

Response: We have added the years in which the ESA and CWA were passed.

Comment 4: Were any correlation tests performed to rule out correlated variables from the GLMMs?

Response: Thank you for the question. We now describe the correlation test used to evaluate collinearity among predictor variables used in linear models at the end of the Methods, and report that these variables did not exhibit evidence of collinearity in the Results.

Comment 5: The authors should provide a table of the variables used in the study. 

Response: All variables used in this study are currently listed in Tables 1 and 3. Tables 1 and 2 include all water quality measures that served as response variables in the different analyses. Table 3 provides AIC weights for each of the landscape condition variables (e.g. percent impervious, percent ag, etc.) that were used as predictor variables. 

Comment 6: Figure 1: Please provide a map of the USA with the highlighted study area.

Response: We have added an inset map of the United States to this figure, which shows the larger study area.

REVIEWER 2

Comment: The introduction is too long and should be shortened and made more concise

Response: We have revisited the introduction and made revisions that have shortened the length substantially. 

Comment: 1). Analysis of biota in contaminated and uncontaminated areas is never explicitly compared, paired tests and analysis are needed to establish a causal impact. The manuscript repeatedly states that tolerances are being exceeded, but the impacts on biota are not clearly analysed. Much of the manuscript reads as if the initial aim was to look at mining and pollution, and that biodiversity was added later to increase readership and relevance.

Response: These comments are helpful for understanding how readers may interpret the purposes and scope of this work, which we address in two parts. 

First, the goals of this research are to test if there are relationships between remotely sensed data (satellite imagery) of MTR to in-stream water quality data for parameters related to wildlife health and conservation, then link any such relationships to the regulatory implications under, especially, the ESA. We believe it is unnecessary and beyond the scope of this work to have to analyze the impacts of contamination on biota for this contribution to be acceptable for publication. These results have been reported in other literature and are used by scientists and regulatory entities like those that implement the ESA. Taken to its logical end, the reviewer's comment would require that any research published not only cite other work but replicate such work before inferences can be drawn. That written, we have revisited the manuscript throughout to ensure that the scope is clear and that potentially causal relationships are conveyed appropriately.

Second, the biodiversity component of the research was first-and-foremost; two of the authors are from a wildlife-focused nonprofit, and the third from one that has a great interest in biodiversity. We have revisited the manuscript with your observation in mind and made edits to help clarify that focus. In the Introduction we make explicit our interest in assessing the potential impacts of mining on imperiled species through its effects on water quality.

Comment: Many more explicit details on the impacts are needed. 

Response: While it is not entirely clear which impacts the reviewer is referencing, following on our response above, we respectfully disagree that this contribution requires many more explicit details of impacts [on biodiversity] to be a contribution that advances science. As noted above, the impacts of the water quality measures we analyze to various taxa have been documented elsewhere, and we reference this literature throughout the manuscript.

Comment: Secondly, and following on this, better species specific analysis (i.e. by guild) are needed in relation to sensitivity, and contaminant specific analysis also needed. The main headline here is "mining contaminates rivers, this could be problematic" but explicit analysis on how species are impacted is needed. This will require extensive restructuring and further analysis to elaborate the findings of the paper.

Response: Related to the comments above, we believe there is a misunderstanding of the goal of the present contribution. Our goal is to test the hypothesis that there is a relationship between MTR changes as detected using satellite imagery and in-stream water quality measurements in the catchments of those MTR footprints for parameters that have known effects on biodiversity, in particular ESA-listed species for which regulations have established thresholds of exposure given the best available scientific information. Our goal is not to do a meta-analysis of the effects of particular contaminants on species or the sensitivity of those species, nor to do in a single undertaking a full causal chain analysis that goes from satellite to organism response. The latter is very interesting and building on this contribution could be a subsequent step in this research program, but for our purposes, the satellite to instream measurements connection, placed in the context of ESA regulations, is sufficient.

JOUNRAL

Comment 1: Please ensure that your manuscript meets PLOS ONE's style requirements, including those for file naming. The PLOS ONE style templates can be found at https://journals.plos.org/plosone/s/file?id=wjVg/PLOSOne_formatting_sample_main_body.pdf and https://journals.plos.org/plosone/s/file?id=ba62/PLOSOne_formatting_sample_title_authors_affiliations.pdf

Response: We have made changes throughout the manuscript to meet the requirements.

Comment 2: We noted in your submission details that a portion of your manuscript may have been presented or published elsewhere. "Yes. The mine footprint data used to estimate was previously published in Pericak A et al. (2018) Mapping the yearly extent of surface coal mining in central appalachia using landsat and Google Earth Engine. PLoS One 13(7). These data are publicly available at: https://skytruth.org/mtr-data-files/" Please clarify whether this publication was peer-reviewed and formally published. If this work was previously peer-reviewed and published, in the cover letter please provide the reason that this work does not constitute dual publication and should be included in the current manuscript. 

Response: The MTR footprint data were the focus of a paper published in PLoS One, as noted in the comment. We do not believe this could possibly be considered dual publication because the previous paper did not connect the MTR data to in-stream water quality data, which is the focus of the present contribution; we simply are using the MTR as an input dataset. We have provided an explanation to this effect in the cover letter.

Comment 3: We note that Figure 1 and Supplementary Figures F1, F2 in your submission contain map/satellite images which may be copyrighted. All PLOS content is published under the Creative Commons Attribution License (CC BY 4.0), which means that the manuscript, images, and Supporting Information files will be freely available online, and any third party is permitted to access, download, copy, distribute, and use these materials in any way, even commercially, with proper attribution. For these reasons, we cannot publish previously copyrighted maps or satellite images created using proprietary data, such as Google software (Google Maps, Street View, and Earth). For more information, see our copyright guidelines: http://journals.plos.org/plosone/s/licenses-and-copyright. We require you to either (1) present written permission from the copyright holder to publish these figures specifically under the CC BY 4.0 license, or (2) remove the figures from your submission: 

3.1. You may seek permission from the original copyright holder of Figure 1 and Supplementary Figures F1, F2 to publish the content specifically under the CC BY 4.0 license. 

3.1. You may seek permission from the original copyright holder of Figure 1 and Supplementary Figures F1, F2 to publish the content specifically under the CC BY 4.0 license. 

3.2. If you are unable to obtain permission from the original copyright holder to publish these figures under the CC BY 4.0 license or if the copyright holder’s requirements are incompatible with the CC BY 4.0 license, please either i) remove the figure or ii) supply a replacement figure that complies with the CC BY 4.0 license. Please check copyright information on all replacement figures and update the figure caption with source information. If applicable, please specify in the figure caption text when a figure is similar but not identical to the original image and is therefore for illustrative purposes only.

Response: We have replaced the basemap imagery in Figure 1 with Landsat data, which are provided publicly by the USGS and NASA. We have added language to the Figure 1 caption providing attribution to this source as recommended by NASA https://landsat.visibleearth.nasa.gov/. We do not intend to publish any supplemental figures. Those referenced, unless we are mistaken, refer to the figures contained in the study previously published in PLoS One whose output data we build upon. This document was included for review purposes to make explicit where data used in this manuscript had previously been published. We have not modified or added new legends for these figures.

Comment 4: Please include captions for your Supporting Information files at the end of your manuscript, and update any in-text citations to match accordingly. Please see our Supporting Information guidelines for more information: http://journals.plos.org/plosone/s/supporting-information. 

Response: We have updated the file name for our supplemental material, as well as all in-text references in accordance with PLoS standards.

---

## [Decision Letter · Decision Letter 1]

21 Jul 2021

PONE-D-20-28542R1

Linking mountaintop removal mining to water quality for imperiled species using satellite data

PLOS ONE

Dear Dr. Evans,

Thank you for submitting your manuscript to PLOS ONE. After careful consideration, we feel that it has merit but does not fully meet PLOS ONE’s publication criteria as it currently stands. Therefore, we invite you to submit a revised version of the manuscript that addresses the points raised during the review process.

Editor's comments.  I think this manuscript is nearly there.  I do suggest that the discussion could be shortened and focussed more closely on the title.  This may then deal with many of the third reviewer's comments

If applicable, we recommend that you deposit your laboratory protocols in protocols.io to enhance the reproducibility of your results. Protocols.io assigns your protocol its own identifier (DOI) so that it can be cited independently in the future. For instructions see: http://journals.plos.org/plosone/s/submission-guidelines#loc-laboratory-protocols. Additionally, PLOS ONE offers an option for publishing peer-reviewed Lab Protocol articles, which describe protocols hosted on protocols.io. Read more information on sharing protocols at https://plos.org/protocols?utm_medium=editorial-emailutm_source=authorlettersutm_campaign=protocols.

We look forward to receiving your revised manuscript.

Kind regards,

Judi Hewitt

Academic Editor

PLOS ONE

Journal Requirements:

Additional Editor Comments (if provided):

Editor's comments. I think this manuscript is nearly there. I do suggest that the discussion could be shortened and focussed more closely on the title. This may then deal with many of the third reviewer's comments

Reviewers' comments:

Reviewer's Responses to Questions

**Comments to the Author**

1. If the authors have adequately addressed your comments raised in a previous round of review and you feel that this manuscript is now acceptable for publication, you may indicate that here to bypass the “Comments to the Author” section, enter your conflict of interest statement in the “Confidential to Editor” section, and submit your "Accept" recommendation.

Reviewer #1: All comments have been addressed

Reviewer #3: (No Response)

2. Is the manuscript technically sound, and do the data support the conclusions?

Reviewer #1: Yes

Reviewer #3: Partly

3. Has the statistical analysis been performed appropriately and rigorously? 

Reviewer #1: N/A

Reviewer #3: No

4. Have the authors made all data underlying the findings in their manuscript fully available?

Reviewer #1: Yes

Reviewer #3: No

5. Is the manuscript presented in an intelligible fashion and written in standard English?

Reviewer #1: Yes

Reviewer #3: No

6. Review Comments to the Author

Reviewer #1: I have checked this paper earlier. The paper is now ready for acceptance. I have no further comment.

Reviewer #3: The water quality data show that chronic and acute thresholds for aquatic life were exceeded thousands of times between 1985 and 2015 in streams that are important to the survival and recovery of species on the Endangered Species List. {which parameters? Name these parameters]

. Linear mixed models showed that levels of conductivity, manganese, sulfate, sulfur, total dissolved solids, total suspended solids, and zinc increased as the proportion of the area draining into a monitoring station that was mined increased. [mention the increase as %]

Introduction:

The release of alkaline mine drainage from weathering of rock and site waste at mine sites elevates conductivity and concentrations of metallic ions that negatively impact aquatic biota [12], leading to decreases in aquatic biodiversity [10,13]

METHODS

This portion is not clear. Be specific, for examples, under “Water Quality data” -- . Additionally, we collected flow rate and temperature data. # where it has been used? And how it has been used?

Table 1. Chronic and acute toxicity thresholds were exceeded many times in waterways important to listed aquatic species [NOT CLEAR?], For examples, Arsenic 39 (chronic) 39 (acute), what are these values? Is that concentration?? Or what? Not Clearly explained in the text.

# Water quality data were provided in different units, and we standardized all measures of concentration to ug/L, temperature to degrees Celsius, conductivity to µS /cm, and turbidity to NTU.

## wright inside the table for better clarity.

For Zinc 451(ug/L) for Acute and 451 (ug/L) for chronic. PLEASE CHECK THE DATA from the sources, where it has been collected? How concentration of Zn (451 ug/L) will be same for chronic and acute toxicity??

RESULTS

From m 0 to 343 km (x = 77.7 km, σ 2 = 4071 km), check data

# The number of observations at each site ranged from 1 to 275. What is the meaning?

Table 2: Total dissolved solids Total

Total dissolved solids Dissolved* (* implies dissolved?)

Total suspended solids Total

Check the solids nomenclature,

CONCLUSIONS

The conclusions must be based on the study undertaken. Entire conclusions discussed impacts of mining on imperiled species through degradation of water quality. It that is so, then in results section provide some tables which are now added as supplementary materials.

Overall comments:

There has been nobility in this approach, and could be useful, but methods results are not clear. It is more of secondary data used in a statistical tools. In such cases, a flow chart should be provided for meaning full interpretation of existing data.

Introduction part is very lengthy, it can be reduced substantially (at least 30-40%)

Methods: section needs through rewriting and Cleary mention commensurate with objectives of study.

I have a strong reservation for recommending this paper for publication, however, this article may be again rewrite and resubmit.

7. PLOS authors have the option to publish the peer review history of their article (what does this mean?). If published, this will include your full peer review and any attached files.

Reviewer #1: **Yes: **Syed Ainul Hussain, Wildlife Institute of India

Reviewer #3: **Yes: **Prof Subodh Kumar Maiti, IIT(ISM) Dhanbad, India

---

## [Author Response · Author response to Decision Letter 1]

14 Sep 2021

Editor's comments. 

I think this manuscript is nearly there. I do suggest that the discussion could be shortened and focussed more closely on the title. This may then deal with many of the third reviewer's comments

We appreciate the positive feedback. In response to this and the reviewers’ suggestions we have shortened both the Introduction and Discussion. Specifically, we narrow and limit the scope of our discussion of conservations laws and regulations, except to make direct connections to the ways in which the results of this analysis directly inform their implementation.

Reviewer #1:

 I have checked this paper earlier. The paper is now ready for acceptance. I have no further comment.

We appreciate the repeated review and help in shaping this manuscript for publication, and are happy to hear that we were able to address the previous suggestions.

Reviewer #3: 

The water quality data show that chronic and acute thresholds for aquatic life were exceeded thousands of times between 1985 and 2015 in streams that are important to the survival and recovery of species on the Endangered Species List. {which parameters? Name these parameters]

As suggested, we have included the names of each of the parameters for which chronic and acute aquatic life thresholds were exceeded: Aluminum; Arsenic; Cadmium; Conductivity; Copper; Lead; Manganese; Mercury; pH; Selenium; and Zinc

Linear mixed models showed that levels of conductivity, manganese, sulfate, sulfur, total dissolved solids, total suspended solids, and zinc increased as the proportion of the area draining into a monitoring station that was mined increased. [mention the increase as %]

We have included the range of beta coefficients for these analyses in this sentence, presenting the average increase in each measure for a one percent increase in mined area. The sentence now reads:

“Linear mixed models showed that levels of conductivity, manganese, sulfate, sulfur, total dissolved solids, total suspended solids, and zinc increased by 6.73E+01 to 3.30E+06 for one percent increase in the mined proportion of the area draining into a monitoring station.”

Introduction:

The release of alkaline mine drainage from weathering of rock and site waste at mine sites elevates conductivity and concentrations of metallic ions that negatively impact aquatic biota [12], leading to decreases in aquatic biodiversity [10,13]

As far as we can tell, this sentence is identical to what is currently in the manuscript. If the reviewer is suggesting any changes it is unclear what they are.

Introduction part is very lengthy, it can be reduced substantially (at least 30-40%)

We have condensed the two Introduction paragraphs describing the legal and regulatory context in this work into a single paragraph. This aligns with a shortening of the Discussion aimed at focusing the text more closely on the results of this study. Additionally, we have made reductions in the length of the Introduction by eliminating and shortening sentences throughout.

Methods: section needs through rewriting and Cleary mention commensurate with objectives of study.

We have edited several sentences that were flagged as confusing or lacking clarity below, as well as re-organized the order in which the Table 1 is presented relative to text describing its contents.

This portion is not clear. Be specific, for examples, under “Water Quality data” -- .

We are not entirely clear as to what this comment pertains, but have re-read the Methods section for clarity, rearranging sections to improve the logical presentation of the data collection and analyses. 

Additionally, we collected flow rate and temperature data. # where it has been used? And how it has been used?

We initially collected these variables for potential use in linear mixed models. We have amended this sentence acknowledging that these variables were not used. Additionally, in the following section we omit describing the conversion of temperature to degrees Celsius.

Table 1. Chronic and acute toxicity thresholds were exceeded many times in waterways important to listed aquatic species [NOT CLEAR?], For examples, Arsenic 39 (chronic) 39 (acute), what are these values? Is that concentration?? Or what? Not Clearly explained in the text.

These values are counts representing the number of times a recorded water quality measure exceeded standard thresholds for aquatic life, and thus are unitless. This description is provided in the Table footer:

“Table shows the number of times that any recorded value (e.g. ‘Dissolved’, ‘Total’, etc.) of each water quality measure exceeded standard thresholds for aquatic life, and the number of different monitoring stations at which these events occurred. These data only consider measures taken from 1,887 monitoring stations whose drainage basin contained a stream that was designated as important to species survival and recovery.

aNo chronic or acute thresholds provided for measure”

Thresholds are not species specific, and we describe their origin in the Methods section as follows:

“To identify acute and chronic toxicity thresholds for different water quality measures, we used the state water quality standards for aquatic life administered by Virginia under the Clean Water Act. These standards are approved by the Environmental Protection Agency and are used to determine waterway impairment requiring mitigation. The water quality standards thus represent an agreed upon set of thresholds necessary to maintain suitability of waterways for aquatic species. The Virginia standards were identical to those from Kentucky, Tennessee and West Virginia.”

We have added a citation for the Vriginia Clean Water Act standards. We have moved the location of Table 1 to follow this description so that the context of the data presented in the table is more clear.

# Water quality data were provided in different units, and we standardized all measures of concentration to ug/L, temperature to degrees Celsius, conductivity to µS /cm, and turbidity to NTU.

## wright inside the table for better clarity.

We have added units to Table 2, which presents regression coefficients estimating the scale of increase or decrease in each water quality measure. As noted above, Table 1 reports the frequency with which thresholds for aquatic life established for each measure were exceeded. Thus, we omit units here.

For Zinc 451(ug/L) for Acute and 451 (ug/L) for chronic. PLEASE CHECK THE DATA from the sources, where it has been collected? How concentration of Zn (451 ug/L) will be same for chronic and acute toxicity??

Assuming these comments refer to Table 1, the reviewer has misunderstood the data being presented in the table, which consists of counts and not concentrations. We have moved the location of the table to the end of the Methods section, to provide more complete context for these data.

RESULTS

From m 0 to 343 km (x = 77.7 km, σ 2 = 4071 km), check data

We have double checked our data and confirm this statement is correct. We have modified how we present the results, as 0 km may cause confusion. We now state that:

“Distances between stations and mines were exponentially distributed, ranging from 1 to 343 km (x = 77.7 km, σ2 = 4,071 km).”

# The number of observations at each site ranged from 1 to 275. What is the meaning?

We have added language clarifying that an observation refers to an occasion on which water quality data was recorded at a site. The sentence now reads:

‘The number of observations (i.e., occasions on which water quality data was recorded) at each site ranged from 1 to 275.’

Table 2: Total dissolved solids Total

Total dissolved solids Dissolved* (* implies dissolved?)

Total suspended solids Total

Check the solids nomenclature,

These are the measures provided by the national water quality data portal, as recorded by the U.S. Geological Survey, National Park Service, and other state and federal agencies. 

Conclusions

The conclusions must be based on the study undertaken. Entire conclusions discussed impacts of mining on imperiled species through degradation of water quality. It that is so, then in results section provide some tables which are now added as supplementary materials.

We have reduced the length of the Discussion by truncating sections less directly related to the study at hand. Specifically we shorten our discussion of SMCRA and the regulatory obligations of the Fish and Wildlife Service under section 7 of the Endangered Species Act. 

We mirror these changes in the Conclusion, although retain a focus on placing the findings of this study in a broader conservation and regulatory context. We have edited language to connect the outcomes of this study more clearly to these broader conclusions.

Overall comments:

There has been nobility in this approach, and could be useful, but methods results are not clear. It is more of secondary data used in a statistical tools. In such cases, a flow chart should be provided for meaning full interpretation of existing data.

Previous reviewers and editors have not had challenges understanding the study design, use or analysis of data. In response to this comment, we have invited a colleague unfamiliar with the project to evaluate the manuscript for clarity and interpretability. They did not identify any major barriers to understanding the methodology and analytical approach, beyond clarifying some of the points regarding units and measures raised above. At this point, we are not inclined to provide a flow chart as an additional figure, and believe the changes made to the Methods and Results are sufficient to provide a clear interpretation of how the data were used and analyzed.

I have a strong reservation for recommending this paper for publication, however, this article may be again rewrite and resubmit.

We appreciate the feedback provided and believe we have been able to address all current and previously raised concerns.

---

## [Editor Report · Decision Letter 2]

14 Oct 2021

Linking mountaintop removal mining to water quality for imperiled species using satellite data

PONE-D-20-28542R2

Dear Dr. Evans,

We’re pleased to inform you that your manuscript has been judged scientifically suitable for publication and will be formally accepted for publication once it meets all outstanding technical requirements.

Kind regards,

Judi Hewitt

Academic Editor

PLOS ONE
---

## [Editor Report · Acceptance letter]

26 Oct 2021

PONE-D-20-28542R2 

Linking mountaintop removal mining to water quality for imperiled species using satellite data 

Dear Dr. Evans:

I'm pleased to inform you that your manuscript has been deemed suitable for publication in PLOS ONE. Congratulations! Your manuscript is now with our production department. 

Kind regards, 

on behalf of

Dr. Judi Hewitt 

Academic Editor

PLOS ONE